# Use of Interval Therapy with Benzodiazepines to Prevent Seizure Recurrence in Stressful Situations

**DOI:** 10.3390/brainsci12050512

**Published:** 2022-04-19

**Authors:** Roy G. Beran

**Affiliations:** 1Neurology Department, Liverpool Hospital, Liverpool, NSW 2170, Australia; roy.beran@unsw.edu.au; 2South Western Clinical School, University of New South Wales, Sydney, NSW 2170, Australia; 3Ingham Institute for Medical Research, South Western Sydney Area Health Service, Sydney, NSW 2170, Australia; 4School of Medicine, Griffith University, Southport, QLD 4215, Australia; 5Faculty of Sociology, Sechenov Moscow First State University, 119991 Moscow, Russia

**Keywords:** interval therapy, seizure prevention, predictive value, anticipatory treatment

## Abstract

Introduction: Antiseizure medications (ASMs) control 70–75% of seizures. Accepting stress as a trigger for seizures, intervention, at the time of predictable stress, should offer a therapeutic option. Mode of intervention: Intervention requires the maintenance of an accurate seizure diary to identify a recurring pattern. With a questioning approach to that diary, the clinician may intervene when stressful provocateurs occur. Benzodiazepines, such as clobazam, initiated prior to the predictable stress, and continued until after it has passed, should be short lived, making serious adverse effects unlikely. Benzodiazepines have a dual benefit, being both anxietolytic and raising the seizure threshold in patients with epilepsy. Discussion: Many patients claim stress provokes their seizures and may not be aware the stress was the provocateur, until after a seizure occurred, leading to a retrospective claim of the relationship. To confirm the relationship, permitting preventative measures, before exposure to provocative factors, often unable to be avoided, requires maintenance and review of a detailed diary. Interval temporary use of benzodiazepines, such as clobazam, or alternatively clonazepam, diazepam or nitrazepam, offers a reasonable response to obviate subsequent seizures, and should be continued, for a brief period, after the risk has abated. Subsequent review of the diary, over a period of repeated exposures to the identified stress, will confirm or refute the therapeutic effect. Conclusion: Judicious use of interval therapy, with one of the benzodiazepines, covering the period of stressful provocation, offers adjunctive treatment of possible refractory epilepsy, based upon the review of the strictly maintained epilepsy/seizure diary.

## 1. Introduction

The mainstay of treatment to prevent seizures for people with epilepsy is the use of ongoing antiseizure medications (ASMs), which control approximately 70–75% of such seizures [1,2]. Despite the very considerable increase in the number and availability of newer ASMs, there has been very little increase in the efficacy of seizure control offered by these newer ASMs [2]. Many patients report that their epileptic seizures occur as the result of being exposed to stressful situations, many of which may be predictable but unavoidable, although human studies which aim to confirm this phenomenon are less convincing [3]. Novakova et al., suggest that stress is likely to exacerbate susceptibility to epileptic seizures in those individuals with epilepsy and that stress may play a role in triggering “spontaneous” seizures [3]. Van Campen et al., suggest that the stress-sensitivity of seizures relates to alteration of the stress response, which could affect neuronal excitability and, as a consequence, trigger seizures [4].

If one accepts that stress is a significant trigger for seizures, it follows that intervention, introduced at or immediately before the time of predictable stress, should offer a further therapeutic option that may act as an adjunct to the standard use of continuous ASM administration. Studies have suggested that patients in whom seizures are provoked by stress may not be aware of the association between the stress and the provoked seizure, until after the seizure has occurred [5]. Without recognition of the cause-and-effect relationship, it obviates the potential for prophylactic intervention prior to the stress occurring [5]. It follows that, to consider interval intervention to pre-empt the effects of exposure to stress, which is thought to provoke seizures, necessitates confirmation that the perceived stressful situation has a direct causal relationship to the occurrence of seizures which occur in conjunction with that stress, and presumably as a consequence thereof.

## 2. Mode of Intervention

Based on the preceding hypothesis, any form of intervention requires a two stepped approach, namely confirmation of the causal relationship evoked by the stress to provoke the subsequent seizures. Having confirmed that relationship, the clinician must devise an effective mode of intervention that can modify the patient’s response to that stressful situation, to protect against experiencing seizures. To do this effectively requires the patient to maintain an accurate seizure diary, which includes both the timing of seizures, and the presence of any possible extraneous circumstances which occur at the approximate time of the seizures, to explore the potential of a cause-and-effect relationship. There is a body of opinion that questions the reliability of seizure diaries [6] and suggests that inherent inaccuracies in these diaries influence inappropriate interpretation of intervention strategies [7]. What this literature does not acknowledge is that there is no indication to suggest that these seizure diaries are any more or less inaccurate, at any given period of time, and hence, there is nothing to suggest that an identified association, between stressful situations and the occurrence of seizures, is to be discounted because of potential inaccuracies that the diaries may evoke.

Review of that diary may allow the therapist the capacity to identify a recurring pattern, within that patient’s life, which accompanies the expression of seizures. This requires a questioning approach to the seizure diary, specifically aimed to identify any possible recurring pattern(s), be it the repeated visit of a person who evokes stress in the patient, a recurrent work-related situation which affects the patient or a physiological phenomenon, such as catamenial expression of seizures, which may relate to hormonal changes. Only then, having identified such recurring pattern, can the clinician consider trying to intervene at the time of such occurrences, if the same can be predicted and follow an identified pattern.

One mode of intervention that can be used is the introduction of benzodiazepines, such as clobazam (Frisium^®^), initiated prior to the exposure to the predictable stressor, and which should be continued until after the provocative situation has resolved and the patient has returned to his/her normal environment. Long-term use of benzodiazepines is prone to the effects of tolerance, requiring increasing dosages of the medication to achieve the identical therapeutic effects, and benzodiazepines generally are recognised to be associated with the development of both dependence and withdrawal seizures [8,9]. Based on the principle that the provocative stressful influences should be short lived and not be present for a prolonged period, it is unlikely that the patient will experience these benzodiazepine side effects as their use, in the form of interval therapy, should be present for only a short period, during which the stress is present, allowing the interval use of this adjunctive approach, in conjunction with the long-term use of standard ASMs, as per the patient’s routine management of his/her epilepsy.

That does not mean that the patient may not experience some of the other unwanted effects of the benzodiazepines, such as fatigue, lethargy, possible confusion or even paradoxical aggressive behaviour [10], that can occur when exposed to benzodiazepines, and it behoves the clinician to be aware of the potential for these to occur, and to prevent their intrusion into the patient’s life, even during the short period(s) of exposure. It follows that these unwanted effects still need to be discussed with the patient, especially as excessive fatigue or irritability may adversely affect the person’s quality of life.

Benzodiazepines have a dual potential benefit for the patient with epilepsy, because they are more than simple anxietolytic, sedative and hypnotic medications. They also have the benefit of raising the seizure threshold in patients with epilepsy [11], acting as an ASM in their own right. Benzodiazepines are used to abort seizures in the management of status epilepticus [12,13], especially using rapidly active formulations, such as midazolam (Versed^®^, amongst other brand names), thereby confirming their potent role in the management of patients with epilepsy. It follows that their use, as interval therapy, to break the cycle of stress provoked seizures, has a strong basis in logic and has proven to be efficacious in this situation, especially if their use is restricted to commencing just before the onset of the identified stressful situation and withdrawing the medication once the provocation has passed [14]. Within this scenario, there is little risk of withdrawal seizures because the exposure to the benzodiazepine is brief. If the worry exists regarding withdrawal seizures, in a vulnerable patient with brittle epilepsy, then the interval use of benzodiazepines can be somewhat extended, and the medications can be withdrawn over a slightly longer period, of perhaps a week, following the exposure to the stressful situation, to obviate this risk which may follow sudden cessation, and hence withdrawal, where withdrawal seizures pose a serious concern.

## 3. Discussion

Stress is a widely reported provocateur for seizures, in susceptible people with epilepsy, although the literature reports a mixed response when examining this as a specific cause to provoke seizures, which is often identified and specifically reported by those who have epilepsy [3,4]. Many patients who claim that stress provokes their seizures when they are involved in vulnerable situations, may not be immediately aware that it was the exposure to stress that was the provocateur, until after the seizure has occurred [5] and they identify the relationship after the fact. This may result in a retrospective claim of the relationship between the stress and the seizure, as perceived by the patient, but this requires more objective confirmation before it can be accepted as a fact. It follows that there are prerequisite demands to confirm this situation, the first step of which, prior to initiating interval preventative measures, designed to overcome such stress related events, is to convincingly identify and confirm the relationship between the provoking occurrence of a stressful situation and the subsequent seizure, as experienced by the patient. Such confirmation will allow the physician to prescribe preventative measures, such as the use of interval benzodiazepines, that can be instituted before exposure to the identified provocative factor, be it the visit of someone who evokes stress in the patient or a physiological occurrence, such as menstruation in catamenial epilepsy [15]. Often the provocateur cannot be avoided, which dictates adopting an alternative approach, such as the use of interval therapy with the introduction of the temporary use of benzodiazepines.

The interval use of benzodiazepines is not new but has not been as well appreciated as the standard, continuous use of ASMs for controlling seizure activity [16]. There are a host of benzodiazepines which may be used in this situation, such as clobazam (Frisium^®^), clonazepam (Rivotril^®^), diazepam (Valium^®^) or nitrazepam (Mogadon^®^), the choice of which is dependent on the treating physician’s preference and any known individual response that the patient may have to any of these medications. It must be appreciated that benzodiazepines remain a potential agent of dependence and addiction, and hence, their use should be properly supervised [17]. It must be emphasised that their use, within this scenario, is for short-term application during a time of stress, rather than long term, as may be applicable when adopted as long-term anxietolytic agents. It follows that the risk of withdrawal problems is markedly reduced because of the short-term administration, which is restricted to the period of exposure to the stressful provocateur, plus a day or two on either side thereof, and the withdrawal is tapered over a day or two. Further consideration, regarding unwanted adverse effects provoked by the benzodiazepines, is limited by the fact that the dosage is kept to a minimum, as the medication essentially is being used in otherwise benzodiazepine naïve individuals, thereby restricting the potential for drug interactions or the possibility of tolerance.

The reason for singling out clobazam is that it has re-emerged as a viable and effective ASM [16,18], and with a relatively long half-life, when compared with other benzodiazepines, should provide better cover for the patient during the period of exposure to the stressful situation. Clobazam is said to bind less to subunits that mediate sedative effects than do other benzodiazepines, and it acts quickly, maintaining a therapeutic effect for a longer duration due to its active metabolite, N-desmethylclobazam [19]. With potentially reduced sedative properties, this potentially overcomes one of the restrictive qualities thought to possibly reduce the use of benzodiazepines in people with epilepsy [1]. Withdrawal seizures were found to be less of a concern with clobazam than may be the case with other benzodiazepines, although judicious tapered withdrawal following the exposure remains the advocated approach [20].

The dosage of clobazam, to be used as interval therapy in stressful situations, is, as is pertinent with most ASMs, a matter of trial and error. It can be started at 10 mg BID, starting one or two days before the relevant exposure and to be continued, as required, for the shortest possible treatment period, as dictated by the duration of the stress. The dosage should be titrated to the level of clinical efficacy, as predicated by the patient’s response, after trialling the medication in the provocative situations.

To confirm any existing relationship between the potential stressful cause and subsequent seizures, it is imperative for the individual who is experiencing the seizures to maintain a strict diary that documents more than just the occurrence of seizures. It must include a comprehensive catalogue of that which was occurring, in that person’s life, at the time of the seizures. A critical review of that diary is required if one is to identify the potential provocateur, following which the preventative regime can be initiated, as has been described. This will entail the judicious introduction of interval therapy, with one of the benzodiazepines (such as clobazam, as is advocated in this overview), prior to the exposure to the identified stress. The maintenance of the diary must continue, with the meticulous ongoing recording of seizures and other experiences, to establish the efficacy of the intervention. Subsequent review of the diary is imperative, to confirm the therapeutic effect, as a result of the introduction of the interval use of the benzodiazepine, which should be continued for a few days after the stressful episode has resolved. Subsequent review of the diary, over a period of exposures to the identified stress, should confirm or refute the therapeutic benefits of this approach.

## 4. Conclusions

Judicious use of interval therapy, with one of the benzodiazepines, such as clobazam, started prior to the time of exposure to the identified provocative stress, which evoked epileptic seizures, and continued until the resolution of the stressful situation, with a further brief period for added protection, offers an alternative adjunctive intervention for the treatment of possible refractory epilepsy, based upon the review of a strict epilepsy/seizure diary.

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
