# Peer review of "Use of Interval Therapy with Benzodiazepines to Prevent Seizure Recurrence in Stressful Situations"

_brainsci, 2022, doi:10.3390/brainsci12050512_

Round 1

Reviewer 1 Report

1. The topic does not fit well with the special issue topic. The subtopic "Novel approaches to prediction and treatment of intractable seizures" might apply, but it is a stretch.
2. This is an opinion piece, with no data.
3. The relationship between external stressors and seizure control probably suggest that stress can cause seizures, but there is substantial residual uncertainty. The paper might usefully add on a discussion on the highly variable reliability of seizure diaries, in which there is some interesting recent data from very long term EEG.
4. The author proposes a diary-based approach to establish causality between stressors and seizure control. The paper might be expanded with references to the quality control literature, and a statistical discussion or simulation. How many episodes of seizures with/without the precipiating stress would be required to have some reasonable certainty that the stressors was in fact associated? There is sufficient literature on "N of 1 trial" to inform this discussion.
5. Sheryl Haut's 2007 paper based on prospective diaries could inform a simulation.
6. There is little discussion on the potential disadvantages of possibly frequent benzodiazepine use. It would be interesting to see a calculated guess, with estimates from the literature, about how many seizures would be avoided per year versus the risk of pharmacological addiction.
7. In summary, this is an interesting idea, but which should be developed more into a full paper based on own data or data from the literature before it is worth publishing.

Useful references:
Blachut B, Hoppe C, Surges R, Elger C, Helmstaedter C. Subjective seizure counts by epilepsy clinical drug trial participants are not reliable. Epilepsy Behav. 2017;67:122–7.
Duun-Henriksen, Epilepsi 2020; 61(9):1805-1817    A new era in electroencephalographic monitoring? Subscalp devices for ultra-long-term recordings
Gabler, Nicole B., Naihua Duan, Sunita Vohra, and Richard L. Kravitz. "N-of-1 trials in the medical literature: a systematic review." Medical care (2011): 761-768.
Mirza RD, Punja S, Vohra S, Guyatt G. The history and development of N-of-1 trials. Journal of the Royal Society of Medicine. 2017 Aug;110(8):330-40.
Uhlmann C, Fröscher W. Low risk of development of substance dependence for barbiturates and clobazam prescribed as antiepileptic drugs: results from a questionnaire study. CNS neuroscience & therapeutics. 2009 Mar;15(1):24-31.

Author Response

  1. I agree with the referee that “The subtopic "Novel approaches to prediction and treatment of intractable seizures"  may provide a better fit.
  2. What was asked of me was to offer something that could be considered useful for patient care and again I agree that what has been offered is by way of an opinion piece but that is often the nature of ‘coal face’ clinical medicine.
  3. I take it that the referee agrees that stress may be a provocateur for seizures but questions the use of seizure diaries. I agree that seizure diaries will only capture those seizures of which the patient is aware and may miss quite a few but this will have been the case both before and after the application of interval therapy with benzodiazepines. It could be argued that, with Benzodiazepines, the patient may be more aware and hence there may well be an improvement in recognition of seizures so an argument that seizure diaries are inaccurate may well be correct but there is nothing to suggest that it is worse with benzodiazepines.
  4. I have acknowledged the referee’s questioning of the accuracy of seizure diaries and have added 2 of the suggested references to the section of the paper under the rubric of “Mode of Intervention”.
  5. As a consequence of adding the additional references I have also altered the numbers of the subsequent references within the text.
  6. The referee makes reference to N of 1 studies, implying that these have value within the context of this submitted paper. What is presented, in this paper, was prepared at the behest of the Journal which was in search of additional material and definitely does not constitute an N of 1 study. I suggest that what is provided, as is suggested by this referee, is a ‘novel approach’ which offers readers a novel capacity ‘to prediction and treatment of intractable seizures’ as has been suggested by this referee and hence a reference to N of 1 studies lacks relevance to the submitted paper.
  7. I am indebted to this referee who stated, “ this is an interesting idea, but which should be developed more into a full paper based on own data or data from the literature”. This is exactly the purpose for publishing this paper which is based on personal experience of having applied the theory and philosophy underpinning the concept within this paper. As a coal face clinician I am providing the ‘interesting idea’ and expect that its acceptance for publication will provide the impetus for others to further develop and explore the application of this approach.

Reviewer 2 Report

The authors report a case about the use of intermittent treatment with benzodiazepines to prevent seizure recurrence in stressful situations. This case has implications for the treatment of patients with seizures. I have no comments.

Author Response

This referee has appreciated the wisdom that underpins the preparation of this paper and has indicated a lack of need to rectify anything contained within the paper. This provides reinforcement for point 6 of the response to referee number 1.

Reviewer 3 Report

In this manuscript, the author described a possible intervention to address stress as a possible triggering factor for seizures in patients with epilepsy.  This manuscript is classified as a Case Report. However, the journal guidelines clearly describe “Case reports present detailed information on the symptoms, signs, diagnosis, treatment (including all types of interventions), and outcomes of an individual patient”, whereas in the case of this manuscript the intervention is not applied to a single patient. Obviously, since the manuscript is a general proposal of an intervention for a large category of patients, the case description is missing. Also, the proposed pharmacological approach is limited to benzodiazepines, but more options are currently available to reduce the stress burden in patients (doi: 10.1002/1348-9585.12243, doi: 10.1016/j.clnesp.2021.07.027, doi: 10.1177/1367493517738123, etc.).

Author Response

  1. Referee 3 suggests that this paper is a case report which is far from the case and I am indebted to Referee 1. Who has acknowledged that it represents a ‘novel approach’ for the prediction and management of intractable seizures, rather than a “case study”.
  2. I have no problem with Referee 3’s comment that there may be alternative approaches to manage stress in patients with epilepsy. In raising this comment, Referee 3 is ignoring the accepted fact that the family of medications, represented by the benzodiazepines, also represents a class of medications known to exhibit antiseizure properties. This paper is not submitted to argue that benzodiazepines are the only antiseizure medications that may be useful to manage stressful situations in patients with epilepsy but rather it has offered a ‘novel approach’ which adopts a class of medication know to have both antiseizure effects as well as being anxietolytic in nature.

Reviewer 4 Report

The article is interesting, however the article provides no data. I don't know the number of patients, the type of study? However the article is not a review article. So, I suggest that the data be presented in table or graphical form.

Author Response

  1. I reiterate that what is provided, within this report, is the canvassing of a ‘novel’ idea, rather than a case report, and should act as a stimulus for others to try what I have personally found to be efficacious but not something which I have subjected to clinical trialling. It offers an idea for others to pursue and to consider as a concept that awaits ‘proof of concept’ on the basis of a trial rather than proving the point. It offers the foundation for future work, rather than being conclusive, although I would stress that it is an approach which I have adopted in my clinical practice with good effect.

Round 2

Reviewer 1 Report

It is an interesting paper, and the idea deserves publishing in some form but I am not certain that this paper is strong enough to explore an idea with clear possible benefits, but also downsides.

The author did not understand my point about n=1 clinical trials. n=1 clinical trials is a one-patient trial, in which the patient is exposed to placebo or active drug for a period of time, and then switched. This type of trial is the gold standard evidence for individual-level patient effect. They are difficult to perform, and are not suitable for curative drugs. However, for epilepsy they are a conceivable way of documenting any reduction in seizure frequency.  They are informally used in many settings. See for example this paper: https://www.ncbi.nlm.nih.gov/pmc/articles/PMC3118090/

My main concern about the paper is that could be used to justify widespread use of clobazam in epilepsy with insufficient evidence. I think it would be publishable if the author provided at least a simulation study, which can be done in Excel or similar and presented in a table format. For example, if 100 patients were subjected to this strategy, what percent are likely to get 50% seizure reduction, or seizure freedom? What percentage could be expected to develop mild, or severe, tolerance? What would be the expected net benefit be? There may well be a net benefit that is currently unexplored. 

In addition, it would be valuable if the author set out some start- and stop-rules. It would seem a definite contraindication if there was recent ongoing substance abuse. When would such therapy be withdrawn? Clobazam is not always easy to stop.  What would the length of a pre-treatment seizure diary have to be in order to have a reasonable estimate of seizure frequency, so that a treatment effect can be estimated?  A statistician could possibly be consulted, as this is a Poission/event count problem.

If a minimum quantitative simulated basis for decision-making based on published data cannot be presented, I think the paper should not be published.

Author Response

Regarding the comments of referee 1, it is unethical to subject patients to a placebo controlled trial when the data are well accepted, as a matter of fact, and the use of placebo control is to subject the patient to something that is clearly ineffective when the consensus is that the alternative treatment provides the patient with protection, as is my appreciation of the current situation with the use of benzodiazepines to modify stressful responses in patients with epilepsy. Thus, to subject the patients to an 'N of 1' trial, as advocated by referee 1, is, at least to my interpretation, unethical and emphasises the need to publish the paper as it brings something to the fore that has been overlooked by, what would appear to be, a sizeable sector of the therapeutic community, as represented by your referees. If referee 1 feels that our experience is dubious then (s)he is at liberty to apply the 'N of 1' test, to determine its validity, as (s)he is not restricted by my knowledge that it works and hence it would not be unethical, within that scenario, for him/her. Placebo controlled trials are of benefit when there is need to prove a concept but, in this situation, as presented in this paper, it is my sincere opinion that the approach is without contention, despite that which has been suggested by referee 1.

I can accept the comments about benzodiazepines being potentially drugs of addiction and abuse and hence the need for some guidelines for their use and have added additional comments plus a further reference to underpin these comments ( see lines 144 - 155).

Reviewer 3 Report

The manuscript was modified as required by reviewers. A further unclear point is how the complex pharmacological therapy on which patients are generally maintained could be reconciled with the proposed intermittent therapy aimed to prevent the stress burden effects. Additionally, could any rebound effect be excluded after the interruption of anxiolytic medication? For instance, a rebound effect related to the GABA-A receptor modulator allopregnanolone was recently observed in rats receiving an antiseizure medication. Could this phenomenon be excluded for the proposed intervention?

Author Response

The comments from referee 3 have been accommodated within the additional inclusion, within the paper, as set out above, in lines 144- 155.  

Reviewer 4 Report

The author should apply this mode of intervention to several patients and obtain scientific proof of its efficiency. Data on the patients where the technique was applied should be presented.

Author Response

As per referee 4, this approach has become axiomatic, at least within my circle of neurologists, and what causes me the most surprise is the incredulity of the referees which suggests that they  have not applied that which is so common within both my practice and that of my colleagues.
My aim, in preparing this paper, was to offer something which I have accepted as standard practice for many years, and to set it out for others to follow, so as to offer the journal the material it sought and to do so at no expense to me.

The referees seemingly unfamiliar with a treatment modality which we have adopted for decades. This alone would justify its publication and, for the sceptics, they are at liberty to trial an 'N of 1' approach which, at least within my practice and experience, would be both unethical and unnecessary.